# Self-Powered Bioelectrochemical Nutrient Recovery for Fertilizer Generation from Human Urine

**Stefano Freguia** [1,]*, **Maddalena E. Logrieco** [2], **Juliette Monetti** [1], **Pablo Ledezma** [1], **Bernardino Virdis** [1] and **Seiya Tsujimura** [3]

[1] Advanced Water Management Centre, The University of Queensland, St. Lucia, QLD 4072, Australia; j.monetti@awmc.uq.edu.au (J.M.); p.ledezma@awmc.uq.edu.au (P.L.); b.virdis@awmc.uq.edu.au (B.V.)

[2] Politecnico di Torino, Corso Duca degli Abruzzi 24, 10129 Torino (TO), Italy; maddalenaeloisa.logrieco@studenti.polito.it

[3] Faculty of Pure and Applied Sciences, University of Tsukuba, 1-1-1 Tennodai, Tsukuba, Ibaraki 305-8573, Japan; seiya@ims.tsukuba.ac.jp

* Correspondence: s.freguia@uq.edu.au; Tel.: +61-7-3346-3221; Fax: +61-7-336-54726

**Abstract:** Nutrient recovery from source-separated human urine has been identified by many as a viable avenue towards the circular economy of nutrients. Moreover, untreated (and partially treated) urine is the main anthropogenic route of environmental discharge of nutrients, most concerning for nitrogen, whose release has exceeded the planet's own self-healing capacity. Urine contains all key macronutrients (N, P, and K) and micronutrients (S, Ca, Mg, and trace metals) needed for plant growth and is, therefore, an excellent fertilizer. However, direct reuse is not recommended in modern society due to the presence of active organic molecules and heavy metals in urine. Many systems have been proposed and tested for nutrient recovery from urine, but none so far has reached technological maturity due to usually high power or chemical requirements or the need for advanced process controls. This work is the proof of concept for the world's first nutrient recovery system that powers itself and does not require any chemicals or process controls. This is a variation of the previously proposed microbial electrochemical Ugold process, where a novel air cathode catalyst active in urine conditions (pH 9, high ammonia) enables in situ generation of electricity in a microbial fuel cell setup, and the simultaneous harvesting of such electricity for the electrodialytic concentration of ionic nutrients into a product stream, which is free of heavy metals. The system was able to sustain electrical current densities around 3 A m$^{-2}$ for over two months while simultaneously upconcentrating N and K by a factor of 1.5–1.7.

**Keywords:** bioelectrochemical system; urine; nutrient recovery; microbial fuel cell (MFC); air cathodes

## 1. Introduction

Nutrient management in the modern world is reaching a tipping point. Nitrogen, phosphorus, and potassium are the three key macronutrients that are required in vast amounts to satisfy the food needs of an increasing world population [1,2]. Most of the nitrogen needs are covered by the Haber–Bosch process, which uses fossil energy to fix atmospheric nitrogen into the ammonia molecule, and can be used as a fertilizer, or more frequently converted to urea or salts before land application. The process itself is known to account for approximately 2% of global energy consumption [3]; hence, it is a major contributor to greenhouse gas release and global warming. The excessive loss of nitrogen from land application is a cause for disruption of natural balances and has recently exceeded its planetary boundary [4], implying that the Earth has passed its natural capacity to deal with anthropogenic nitrogen release.

The phosphorus needed for fertilizers is produced from minerals locked in phosphate rock mines, of which there are limited stocks that are expected to reach critically concerning low levels within a century [1,5].

The only way to maintain food security without further harming the planet is to move to new paradigms in urban water management [6] with more sustainable technology applications [7] that can lead to a circular economy of nutrients, whereby nutrients are recovered from waste streams and recycled for fertilizer production [8]. Among the various waste streams generated by modern society, urine has attracted particular attention, as it is a concentrated source of N, P, K, and micronutrients, containing 80% of N and 50% of P discharged with human waste, while accounting for less than 1% of the total wastewater flow in urban systems [9].

Several technologies have been proposed for nutrient recovery from urine, including struvite precipitation [10–13], ammonia stripping [14–17], stabilization + distillation [18–20], electrochemical concentration such as electrodialysis [21,22], and membrane distillation [23,24]. All these technologies require one or more of the following: high energy consumption, high chemical requirements, and/or control systems to ensure operability. For example, the VUNA process utilizes > 70 kWh/kgN as electrical energy to run aeration and distillation units; struvite precipitation requires addition of equimolar MgO to phosphate and uses process controls to maintain optimal conditions for precipitation. Electrodialysis and other membrane-based processes concentrate nutrients and have been shown to effectively reject a range of micropollutants [22,25], but at the price of > 5 kWh/kgN electricity consumption. Microbial electro-concentration was introduced as an alternative to electrodialysis to reduce the energy requirements and eliminate chemical needs [26]. The process, also known as Ugold, still requires ~5 kWh/kgN to drive the electrochemical system, in addition to an aeration system for the removal of organics and recirculation pumps to provide mixing. Moreover, it requires a control system to maintain the anolyte pH within an optimal range.

In this work, we have tested a variation of the process of microbial electro-concentration of urine, where the substitution of a hydrogen-producing cathode with a breathing air cathode with a suitable catalytic layer enabled the operation of the system at short circuit (i.e., at zero cell voltage, hence, at zero power input). The elimination of recirculation pumps was also included to truly attain the world's first self-powered nutrient recovery system. The process is also free from chemical addition and process controls, making it suitable for deployment in off -grid and remote areas.

## 2. Materials and Methods

### 2.1. Hydrolysed Human Urine

Human urine was collected in the male toilets of the Advanced Water Management Centre (University of Queensland, Brisbane, QLD, Australia) using 15 L drums fitted with a large funnel that acted as a waterless urinal. Once the drums contained 10 L of urine, they were swapped for empty ones, and the collected fresh urine was left to hydrolyze in the drums for 3 to 7 d through the action of naturally occurring ureolytic bacteria. After the hydrolysis step, urea was fully converted into ammonia, and Ca and Mg salts precipitated to the bottom of the drum. The supernatant was used as a feed to the microbial electro-concentration cells. Its average characteristics over the experimental period were as follows: pH 8.8; ionic conductivity (EC) 45 mS cm$^{-1}$; 9.5 g L$^{-1}$ total chemical oxygen demand (tCOD), of which 4.4 g L$^{-1}$ was acetic acid; 7.8 g L$^{-1}$ NH$_4$-N; 0.33 g L$^{-1}$ PO$_4$-P; 2.4 g L$^{-1}$ K$^+$; 2.9 g L$^{-1}$ Na$^+$; 0.65 g L$^{-1}$ SO$_4$-S; and 10 mg L$^{-1}$ Ca$^{2+}$. Collection, use, and analysis of human urine were carried out in accordance with human research ethics approval granted by the University of Queensland's Sub-Committee for Human Research Ethics, approval number 2019000024.

### 2.2. Electrochemical Cell Design and Operation

The configuration of the duplicates used in the experiments was a hybrid between an electrodialysis system and a microbial fuel cell, previously described in Ledezma et al. [26]. It consisted of a plate

and frame configuration, where the anode and the air cathode chambers were separated by a product chamber, each with a volume of 200 cm$^3$ (5 cm (width) × 20 cm (height) × 2 cm (depth)). A cation exchange membrane (CEM, CMI-7000, Membranes International, USA) was placed in between the anode and the product chamber, and an anion exchange membrane (AEM, AMI-7000, Membranes International, USA) separated the product chamber from the cathode chamber. Both the membranes had a surface area of 100 cm$^2$, and because of their tendency to deform upon wetting, the product chamber was filled with glass beads to hold the membranes in place. The anodic chamber was filled with graphite granules 2–5 mm diameter (El Carb-100, Graphite Sales, USA) that served as the electrode and biofilm support. Both the glass beads and graphite granules were previously washed sequentially in 1 M HCl and 1 M NaOH, to ensure the removal of trace metals and contaminants [27]. Two graphite rods (5 mm diameter, 9.5 cm length, Morgan Advanced Materials, UK) were used as current collectors at the anode. The rods were inserted into the graphite bed via fittings. A Ag/AgCl (sat. KCl) reference electrode (−0.197 V vs standard hydrogen electrode, SHE) was inserted into a side opening of the anode frame.

The cathode compartment was closed by a frame with an opening of 5 × 20 cm. The air cathode (see next section) had a plastic mesh for mechanical support, and a titanium mesh was used as current collector. The seal was ensured by rubber gaskets sandwiched between each plate/membrane.

Pre-hydrolyzed human urine was fed with a peristaltic pump to the bottom of the anode chamber at a constant flow rate of 0.5 L d$^{-1}$. In the anode, an electrochemically active biofilm was allowed to develop and acclimate to enable the conversion of urine organics to $CO_2$, protons, and electrons. The overflow from the anode compartment was sent through a tube to the cathode inlet, allowing the anode and cathode to operate as hydraulic series. The cathode effluent overflowed into a receiving bottle. No recirculation was applied to eliminate the associated electricity consumption. A peristaltic pump was used for the feed, but in real applications the hydrolyzed urine would be gravity fed. The process is schematized in Figure 1. The product chamber received ions via migration through the membranes. Water flowing into the product chamber, due to osmotic and electro-osmotic processes, generated a flow out of the product chamber. The system was operated for approximately 70 d.

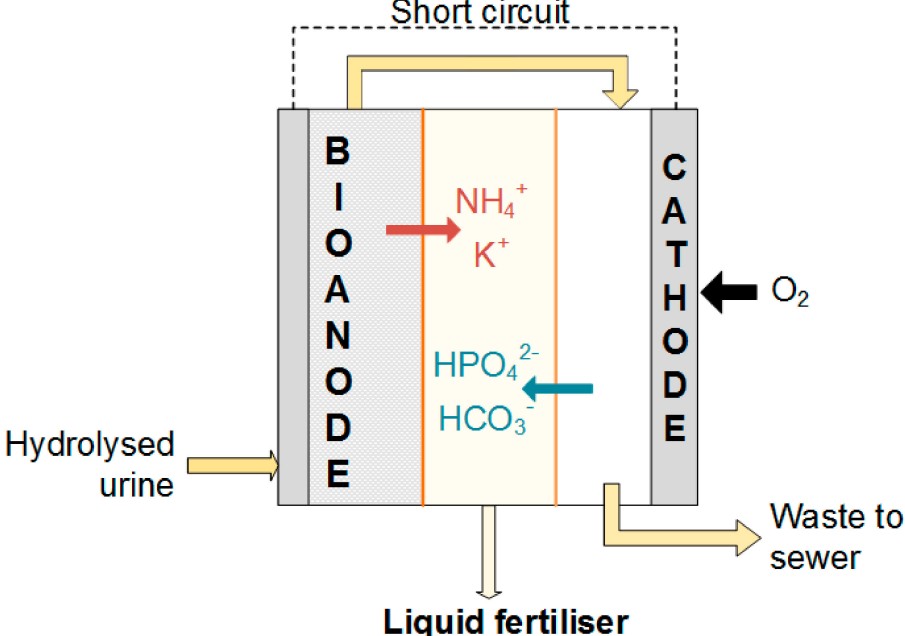

**Figure 1.** Schematic of the bioelectrochemical nutrient recovery system. The process generates its own electricity to drive ionic species (mainly nutrients) into a product chamber, generating a liquid fertilizer.

With the exception of the start-up phase, the electrical operation was a simple short circuit connection between anode and cathode, obtained by applying a voltage of 0 V between anode and cathode via a potentiostat (VMP3, Biologic, France), which also continuously recorded the current. For consistency with previous work, the current densities are reported here as normalized to the cathode/membrane cross-sectional area (i.e., 100 $cm^2$).

## 2.3. Air Cathode

The air cathode consisted of a $20 \times 5$ cm carbon cloth (Fuel Cell Store, USA), coated on the inner side with a catalyst and on the outer side with a gas diffusion layer [28]. The carbon-based catalyst (CC) was synthesized according to Nabae et al. [29,30]. Pyromellitic acid dianhydride (TCI, Japan) and 1,3,5-tris(4-aminophenyl)benzene (TCI, Japan) were briefly polymerized with Fe(acac)$_3$ at 0 °C in acetone. The multistep pyrolysis of the polymer was done at 600 $^0$C for 5 h under $N_2$ and 800 °C for 1 h under $NH_3$ atmosphere. The air cathode diffusion layer was formed by 4 layers of poly-tetrafluoroethylene (PTFE) as a compromise between hydrophobicity and oxygen permeability. An aqueous suspension of PTFE (Sigma-Aldrich) was applied on the carbon cloth with a brush and heated in an electric furnace at 370 $^0$C for 10 min. After 15 min of cooling to return to room temperature, the PTFE coating was repeated 3 times (in total four PTFE layers). PTFE flakes (type 6J, Chemours-Mitsui Fluoroproducts, Japan) were dispersed in isopropanol with a tip-type ultrasonicator. A mixture of CC powder and carbon black was added into the PTFE suspension and dispersed again with the ultrasonicator to obtain a viscous ink. The ink was applied to the opposite side of the PTFE layer of the carbon cloth with a paint brush and dried overnight at room temperature. As a positive control test, commercially available, Pt-loaded carbon (HISPEC 4100; Alfa-Aesar, Australia) was also coated on the carbon cloth. The cathode's catalytic performance was tested in an artificial urine medium ([26], 4.12 g $L^{-1}$ potassium hydrogen phosphate ($K_2HPO_4$), 2.37 g $L^{-1}$ sodium sulphate ($Na_2SO_4$), 0.85 g $L^{-1}$ magnesium chloride hexahydrate ($MgCl_2$–$6H_2O$), 0.38 g $L^{-1}$ calcium chloride ($CaCl_2$), 0.29 g $L^{-1}$ potassium chloride (KCl), 10.79 g $L^{-1}$ ammonium acetate ($NH_4CH_3COO$), 22.14 g $L^{-1}$ ammonium bicarbonate ($NH_4HCO_3$), 6.72 g $L^{-1}$ sodium hydroxide (NaOH), and 1 mL trace elements) by cyclic voltammetry, done with a potentiostat (VMP-3, Biologic, France) at a scan rate of 5 mV $s^{-1}$, in the potential range −0.2 to +0.5 V versus SHE.

## 2.4. Start-Up

The inoculum was a pre-enriched alkaliphilic, ammonia-tolerant, and electroactive consortium obtained from the cathodic effluent of similar reactors originally started from three sources: an acetate-fed bioanode, a urinal, and anaerobic digester sludge [26]. The inoculation was carried out by filling the anode and cathode compartments of the electrochemical cells completely with this inoculum. During the start-up phase (10 d), the anode potential was controlled to 0 V versus SHE, and the anode liquid was recirculated with a peristaltic pump at 0.2 L $min^{-1}$ to allow for a faster development of the electroactive biofilm. After this phase, the anode potential control was stopped, and the system was operated in short circuit mode (zero cell voltage). The recirculation pump was switched off for the rest of the operation.

## 2.5. Chemical Analysis

Liquid samples were taken during the steady-state period (40–65 d since inoculation) from the feed, anode effluent, cathode effluent, and concentrate. Samples were analyzed for $NH_4$-N and $PO_4$-P by flow injection analysis (FIA) and for K, Na, Ca, Mg, S, and metals by inductively coupled plasma optical emission spectroscopy (ICP-OES). Acetic acid was analyzed by high-performance liquid chromatography (HPLC).

## 3. Results and Discussion

### 3.1. Cathode Catalytic Performance

The cyclic voltammograms shown in Figure 2 show that the catalytic layer enabled the reduction of the activation overpotential for the oxygen reduction reaction (ORR), generating a catalytic current with an onset at approximately 0.3 V versus SHE. This improvement, compared to a catalyst-free control, enables power-free electricity production in the system. Voltammograms were recorded with and without a catalyst. In the absence of a catalyst, the ORR current increased at 0.0 V versus SHE (dotted curve in Figure 2). In this case, it is impossible to achieve a self-powered reactor, as the onset potential for oxygen reduction is the same as the anode potential previously used for these systems, and it is in the range of −0.2 to 0 V versus SHE [26,31], thus preventing the generation of significant current densities. When Pt-loaded carbon was used as a catalyst, the onset potential shifted positively by 0.25 V (dashed curve in Figure 2), and 60 A m$^{-2}$ of ORR current could be obtained even at 0.0 V. However, Pt can easily lose its activity by the sulfide (HS$^{-}$) generated from the process of microbial urine decomposition [32]; hence, its use would not be sustainable. Therefore, in this study, we introduced a carbon-based ORR catalyst (CC). Even though, to date, several types of alternative Pt catalysts have been reported, the CC used in this study is known to exhibit the highest catalytic activity as a cathodic catalyst for polymer electrolyte fuel cells [29,30]. This is the first time this catalyst has been used for microbial fuel cell or microbial electrochemical cell applications. The solid black curve in Figure 2 shows the ORR on the CC-modified electrode. The onset potential for ORR was 0.34 V, almost 0.1 V more positive than that observed on the Pt electrode. Importantly, the catalyst worked well in artificial urine and did not show loss in performance during the reactor's operation. Moreover, leakage of electrolyte from the four PTFE layers was not observed.

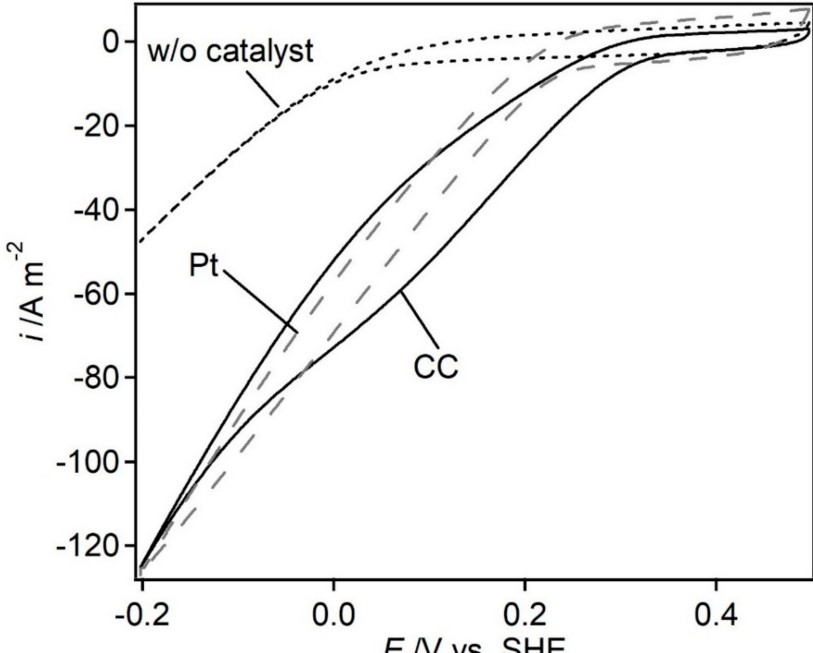

**Figure 2.** Cyclic voltammograms on air cathodes with 4 PTFE layers with different reaction layers: carbon black without catalyst (dotted curve), carbon black with 10% of Pt (dashed curve), and carbon black with 10% CC (solid curve). Electrolyte: artificial urine ([26]), scan rate: 5 mV s$^{-1}$, surface area: 0.5 cm$^2$, room temperature.

### 3.2. Electricity Generation at Zero Power Input

The duplicate reactors were started up with the anode inoculation process described above and with an anode potential of 0 V versus SHE. Electrical current developed within 10 d to over 14 A m$^{-2}$, and still exponentially grew according to the profile shown in Figure 3 for one of the duplicates. After 10 d, short circuit conditions (zero cell voltage) were established, and recirculation was stopped. Current production, which stabilized over 2 months at ~3 A m$^{-2}$, was observed in both reactors. The decrease in current observed after the start-up phase (10 d) occurred as the recirculation stopped, which indicates that recirculation plays an essential role in boosting performance in microbial electrochemical systems due to the mildening of mass transfer limitations. However, recirculation often entails high energy requirements, as described in Ward et al. [33], who estimated that a power of 5 kWh/kgN is required to recirculate the concentrate and diluate streams in an electrodialysis system for nutrient recovery from sludge centrate. While the currents attained in this work without recirculation were one order of magnitude lower than those previously obtained in the Ugold system [26], confirming the importance of hydrodynamics and recirculation in MFC performance [34], this is the first demonstration that this level of current can be achieved in a microbial electro-concentration cell without any power input in the form of electrical voltage or pumping for recirculation (pumping for feeding would be replaced by gravity in real applications). Another remarkable aspect of the electroactive biofilm established at the granular anode is that it was effective at the natural pH of the fed hydrolyzed urine, which averaged 8.8 in this work and was not controlled by chemical dosing.

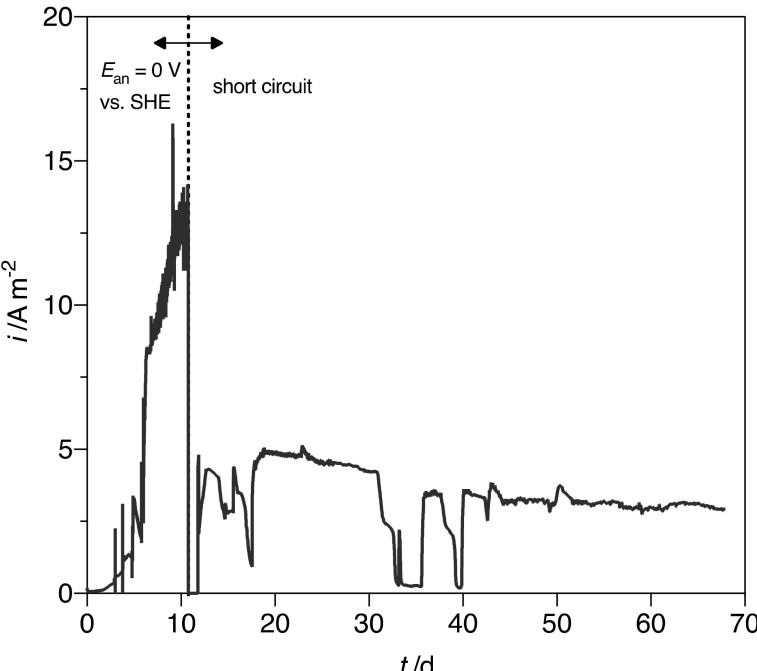

**Figure 3.** Electrical current profile of one of the duplicate reactors. Inoculation occurred at time zero. For 10 d, the system ran at controlled anode potential (0 V vs. SHE), followed by short circuiting the anode and cathode. Occasional current drops were due to power cuts or feed failures.

### 3.3. Nutrient Recovery and Fertiliser Production

The ion exchange membranes in the system acted as sieves to selectively allow small ionic molecules to migrate into the product chamber. The valuable ions being transported are the nutrients $NH_4^+$, $K^+$, $PO_4^{3-}$, $SO_4^{2-}$ and acetate, which is recognized as a valuable organic supplement in fertilizers. The main competing ions are $Na^+$, $Cl^-$ and $HCO_3^-$. Table 1 reveals that N and K accumulated in the product chamber at slightly higher upconcentration factors than Na, which is a useful outcome for the reuse of the product as fertilizer, where large concentrations of Na are undesirable. Only approximately 5% of

the influent flow passed through the membranes (leading to recoveries of N and K around 7%–8%), indicating that the feed flow was too high for the electrical performance of the system. Further work is warranted to optimize the feed rate to enable higher fertilizer recovery. Unlike other ions, phosphate appeared to be down-concentrated in the product, possibly as a reflection of its higher charge (at pH 8.8, $HPO_4^{2-}$ is the prevalent species) and consequently larger hydration sphere. Precipitation of phosphate could also have occurred in the concentrate chamber, as previously described [35]. The analysis shows that heavy metals, including Cd, Cr, Cu, Pb, and Zn, are below detection limits, indicating that they are not upconcentrated from the source-separated urine.

**Table 1.** Performance parameters of the duplicate electrochemical reactors.

|  | Reactor 1 | Reactor 2 |
|---|---|---|
| Average current density | 3.1 | 2.7 |
| N-removal rate (g N m$^{-3}$ d$^{-1}$)$^{(1)}$ | 496 | 409 |
| N-removal rate (g N m$^{-2}$ d$^{-1}$)$^{(2)}$ | 30 | 25 |
| Energy input | 0 | 0 |
| N upconcentration factor | 1.5 | 1.4 |
| P upconcentration factor | 0.6 | 0.6 |
| K upconcentration factor | 1.7 | 1.6 |
| Na upconcentration factor | 1.4 | 1.4 |
| Product-to-feed ratio | 0.054 | 0.046 |

The microbial electro-concentration system hereby described works as a sieve that converts urine into a fertilizer by retaining nutrients while rejecting heavy metals and complex organic chemicals present in urine, as also shown previously [15]. The composition of the product stream, as detailed in Table 2, is aligned with a number of liquid fertilizers currently in use in hydroponic systems and horticulture. This study demonstrates the first self-powered, chemical-free, and control free system for the production of fertilizer from urine.

(1)　　Normalized to total reactor volume
(2)　　Normalized to membrane area (reactor cross-section)

**Table 2.** Composition of the produced fertilizer.

|  | Average | Standard Deviation (n = 6) |
|---|---|---|
| pH | 8.1 | |
| Ionic conductivity, mS cm$^{-1}$ | 77 | |
| NH$_4$-N, mg L$^{-1}$ | 11,600 | 600 |
| PO$_4$-P, mg L$^{-1}$ | 202 | 4 |
| K, mg L$^{-1}$ | 3900 | 200 |
| Na, mg L$^{-1}$ | 4000 | 200 |
| Ca, mg L$^{-1}$ | 1 | 1 |
| Mg, mg L$^{-1}$ | < 1 | |
| SO$_4$-S, mg L$^{-1}$ | 1020 | 70 |
| Acetate, mg L$^{-1}$ | 19,000 | 600 |
| Cd, Cr, Cu, Pb, and Zn, mg L$^{-1}$ | < 1 | |

## 4. Conclusions

This work is the proof of concept of the world's first self-powered, chemical-free, and control-free nutrient recovery system, designed to recover nutrients from human urine in a liquid stream with fertilizer properties. The process is essentially an electrodialysis system embedded in a microbial fuel cell, where the electricity generated by the oxidation of urine organics and the reduction of oxygen on a urine-resistant cathode is used to allow ionic nutrients to migrate into a product stream. The system

attained a sustainable electrical current density of 3 A m$^{-2}$ when run on real hydrolyzed human urine. This level of current has not previously been achieved in the absence of any recirculation or mixing. This electricity enabled the production of a liquid fertilizer with 1.2% N, 0.4% K, 0.02% P, 0.1% S, and no heavy metals. Due to its simplicity and self-reliance, this system could play a role in the uptake of a circular nutrient economy both in high-tech urban and off-grid rural contexts.

**Author Contributions:** Conceptualization, S.F. and S.T.; methodology, S.F., M.E.L., and S.T; software, J.M. and B.V.; formal analysis, S.F., M.E.L, S.T., and J.M; investigation, M.E.L; resources, S.F. and P.L.; data curation, J.M., B.V. and S.T.; writing—original draft preparation, M.E.L. and S.F.; writing—review and editing, S.F., J.M., B.V., S.T., and P.L.; visualization, J.M., B.V., and S.T..; supervision, S.F.; project administration, S.F.; funding acquisition, S.F.

**Funding:** This research was funded by the Australian Research Council, Queensland Urban Utilities and Memtech, grant number LP100200112.

**Acknowledgments:** The authors would like to acknowledge the Australian Research Council, Queensland Urban Utilities, and Memtech (formerly ABR Process) for the funding received through ARC Linkage grant LP100200112. The authors also thank Dr Yuta Nabae at the Tokyo Institute of Technology, Japan for the production method of the carbon-based catalyst.

**Conflicts of Interest:** The authors declare no conflicts of interest. The funders had no role in the design of the study; in the collection, analyses, or interpretation of data; in the writing of the manuscript, or in the decision to publish the results.

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
