# Peer review of "Self-Powered Bioelectrochemical Nutrient Recovery for Fertilizer Generation from Human Urine"

_sustainability, doi:10.3390/su11195490_

Round 1

Reviewer 1 Report

The manuscript deals with issues which had received several times some attention.

However, it is just an experiment, not two much original research work suggestion novel methodology and approaches.

The authors are laud with self-appreciation:

“This work is the proof of concept of the world’s first self-powered, …”

State of the art has not  been adequately detailed and the quality downgraded by multiple references, which are not really analysing the issue.

They should be for each reference several words as a minimum to assess and justify the reference.

The especially previously attempted case should be searched for seriously and more deeply analysed.

These should also be added some economic analysis if the case should be considered more for a realistic development. It is interesting that something might work, however much more important is that it has got a chance it real-life conditions.

For the text clarity would you refrain from using additional words, mostly meaningless filler words, which can be omitted or some archaic words see e.g. “respectively”, “thus”, “hence”, therefore”, “furthermore”, “thereby”, “basically,”, “meanwhile”,” wherein”, “herein”, “hitherto”, “Nonetheless”, “Perceivably” , “whereas”,etc. ?

If the suggested improvements are properly addressed the manuscript could be an interesting contribution.

Author Response

Reviewer 1

The manuscript deals with issues which had received several times some attention.

However, it is just an experiment, not two much original research work suggestion novel methodology and approaches.

We thank the reviewer for pointing out the novel nature of our methodology.

The authors are laud with self-appreciation:

“This work is the proof of concept of the world’s first self-powered, …”

We politely disagree with the judgement that this statement is self-appreciation. We believe this is a fact.

State of the art has not  been adequately detailed and the quality downgraded by multiple references, which are not really analysing the issue.

There seems to be a contradiction in this comment: the reviewer suggests that the literature review is incomplete, yet he criticises that there are too many references. We believe we have covered all the relevant literature.

They should be for each reference several words as a minimum to assess and justify the reference.

The especially previously attempted case should be searched for seriously and more deeply analysed.

It is unclear what the reviewers refers to with these sentences.

These should also be added some economic analysis if the case should be considered more for a realistic development. It is interesting that something might work, however much more important is that it has got a chance it real-life conditions.

This is a proof of concept. As such it is a demonstration at lab scale, and economic assessment would be premature and likely leading to gross errors. We would like to first scale up the technology and then attempt a full techno-economic assessment: this is warranted in a future study.

For the text clarity would you refrain from using additional words, mostly meaningless filler words, which can be omitted or some archaic words see e.g. “respectively”, “thus”, “hence”, therefore”, “furthermore”, “thereby”, “basically,”, “meanwhile”,” wherein”, “herein”, “hitherto”, “Nonetheless”, “Perceivably” , “whereas”,etc. ?

We politely disagree with this statement. Many of these words are commonly used in everyday English, and certainly in written scientific English.

If the suggested improvements are properly addressed the manuscript could be an interesting contribution.

We thank the reviewer for the generally positive evaluation.

Reviewer 2 Report

In the introduction, refer to existing BES-based technologies for nutrient recovery as an extension to the other recovery technologies. N recovery yield, energy consumed and/or recovered during recovery experiments, efficiency of other parameters (such as N recovered for membrane surface/time). Justify why the architecture of the BES reactor of the present study is an advance respect the current state-of-the art. Why the MFC was operated in short-circuit condition and not at low resistance? During acclimation of electro active biofilm, any tests (such as open circuit potential and/or cyclic voltammetry) were carried out to determine biofilm performance? Why not following conductivity in the three chambers over time? As an indirect measure of ions transport. The reactor architecture employed for avoiding recirculation is elegant, and with potential scaling-up applications. The current density achieved during experiments are, as authors state, really high compared to current state of the art. Which is a nice step-forward for the use of BES for nutrients recovery. Did authors asses the changes in AEM and CEM performance after experiment? Changes in perm selectivity and electric resistivity due to (bio)fouling) could play an important role fir nutrients recovery over time.

Author Response

Reviewer 2

In the introduction, refer to existing BES-based technologies for nutrient recovery as an extension to the other recovery technologies.

We have now linked the two paragraphs, using the link “Microbial electro-concentration was introduced as an alternative to electrodialysis…”, at lines 75-76.

N recovery yield, energy consumed and/or recovered during recovery experiments, efficiency of other parameters (such as N recovered for membrane surface/time).

As pointed out below for the response to Reviewer 3, most of these requested parameters are already included in the manuscript. We have now added N recovery normalised to membrane surface to Table 1, including a note to explain the normalisation.

Justify why the architecture of the BES reactor of the present study is an advance respect the current state-of-the art.

We would like to point out that the advance here is not the architecture, but the electrical operation at short circuit with a catalytic air breathing cathode. This step forward is described in details in the Introduction (lines 81-87) and in the Materials and Methods (lines 143-148).

Why the MFC was operated in short-circuit condition and not at low resistance?

This was done to maximise the current density, and therefore the ammonium transfer to the concentrate chamber. There would be only trivial benefit in producing electricity as a by-product.

During acclimation of electro active biofilm, any tests (such as open circuit potential and/or cyclic voltammetry) were carried out to determine biofilm performance?

We have covered the details of the anodic biofilm and its performance (microbial community and cyclic voltametric response) in previous publications (Ledezma et al., FEMS Microbiology Letters, 2018, 365(2), and Ledezma et al., ES&T Letters, 2017, 4(3), 119) where we were using power to drive the process with a hydrogen-producing cathode. The novelty and the focus of this work is on the first successful power-free operation of the system.

Why not following conductivity in the three chambers over time? As an indirect measure of ions transport.

We have monitored ionic conductivity in the chambers. As the system was operated in continuous mode and steady state, the values were roughly constant over time, hence profiles are not worth reporting. The conductivity of the feed is reported at line 100, the conductivity of the concentrate is in Table 2.

The reactor architecture employed for avoiding recirculation is elegant, and with potential scaling-up applications.

We are thankful for such a positive comment.

The current density achieved during experiments are, as authors state, really high compared to current state of the art. Which is a nice step-forward for the use of BES for nutrients recovery.

We thank again the reviewer for the supportive statement.

Did authors asses the changes in AEM and CEM performance after experiment? Changes in perm selectivity and electric resistivity due to (bio)fouling) could play an important role fir nutrients recovery over time.

This is the topic of another student’s project, and was published in Monetti et al., 2019, ACS Omega, 4(1), 2152. We can summarise here that the AEM and CEM maintain their properties, as long as care is taken to separate the Ca and Mg precipitates before feeding.

Reviewer 3 Report

In the manuscript “Self-powered bioelectrochemical nutrient recovery for fertilizer generation from human urine” the authors describe the use of a bioelectrochemical system to convert urine to a nutrient rich fertilizer. In this work the authors describe the set-up used and quantify the ions and nutrients existent in the system, establishing a proof-of-concept of a self-powered, chemical-free and control-free nutrient recovery system.

I consider that the manuscript is important for the microbial electrochemical community, and should be published, although the use of urine as a substrate in a bioelectrochemical system and its nutrient recovery is not a novelty (e.g. Gao et al 2018 Environ. Sci.:Water Res Technol, Cid et al (2018) J. Power Sources).

However, the lack of information regarding active organic molecules, ammonia, tCOD and heavy metals for the beginning and end of the experiment, precludes the estimation of the efficiency of the process. I consider that the missing values are crucial to demonstrate the conclusions made by the authors, that shows that the liquid stream obtained from urine can be used as a fertilizer. Furthermore, the authors should also provide information of the composition of conventional fertilizers, to make it clear the efficiency of the process. I suggest the authors to include in table 2 the information of the composition of the media prior the experiment, and also include the values of conventional fertilizers. Moreover, the amount of acetate should also be included in this table.

For these reasons I consider that for the manuscript to be accepted for publication, the authors should provide this information, and should also discuss the efficiency of the process and compare it with other nutrient recovery processes from urine.

Did the authors identify the electroactive organisms present in the bioreactors? This would also be relevant to identify the organisms that are responsible for the consumption and conversion of nutrients from urine.

Author Response

Reviewer 3

In the manuscript “Self-powered bioelectrochemical nutrient recovery for fertilizer generation from human urine” the authors describe the use of a bioelectrochemical system to convert urine to a nutrient rich fertilizer. In this work the authors describe the set-up used and quantify the ions and nutrients existent in the system, establishing a proof-of-concept of a self-powered, chemical-free and control-free nutrient recovery system.

I consider that the manuscript is important for the microbial electrochemical community, and should be published, although the use of urine as a substrate in a bioelectrochemical system and its nutrient recovery is not a novelty (e.g. Gao et al 2018 Environ. Sci.:Water Res Technol, Cid et al (2018) J. Power Sources).

We agree with the reviewer. The novelty of this work is the technology design.

However, the lack of information regarding active organic molecules, ammonia, tCOD and heavy metals for the beginning and end of the experiment, precludes the estimation of the efficiency of the process. I consider that the missing values are crucial to demonstrate the conclusions made by the authors, that shows that the liquid stream obtained from urine can be used as a fertilizer.

The hydrolysed urine stream has been characterised and it is described in the first paragraph of the Materials and Methods section (lines 99-102). This includes COD, ammonia, salinity, sulfate, sodium, phosphate, and now also acetate. The non-VFA COD in urine is a complex mix of chemicals that are extensively described in the literature, hence we did not characterise it further. The removal efficiency of N and P is given at line 268. Therefore, we believe there is sufficient information in the manuscript to quantify the efficiency of the process.

Furthermore, the authors should also provide information of the composition of conventional fertilizers, to make it clear the efficiency of the process. I suggest the authors to include in table 2 the information of the composition of the media prior the experiment, and also include the values of conventional fertilizers. Moreover, the amount of acetate should also be included in this table.

There are just too many “conventional” liquid fertilisers, and they widely vary in concentration and nutrient ratios. As we do not want to advertise one particular brand, we prefer to avoid including information on commercial materials. As mentioned above, the composition of hydrolysed urine is already given in the text (lines 99-102). We have now included the acetate concentration of the hydrolysed urine (line 101) and in the concentrate (Table 2).

For these reasons I consider that for the manuscript to be accepted for publication, the authors should provide this information, and should also discuss the efficiency of the process and compare it with other nutrient recovery processes from urine.

As discussed above, the efficiency of the process is already included. The message of this work is the nil power consumption required to recover nitrogen. This comparison is already included as the existing processes are detailed at lines 55-80 of the Introduction.

Did the authors identify the electroactive organisms present in the bioreactors? This would also be relevant to identify the organisms that are responsible for the consumption and conversion of nutrients from urine.

We have identified organisms in other similar reactors (same anodes, but with power-draining cathodes) and published the work elsewhere (Ledezma et al., FEMS Microbiology Letters 365(2) 2018). Microbial ecology is not in the scope of this study.

Round 2

Reviewer 1 Report

as the authors rejected most points of the previous review I have no other option than to stick with my previous comments as they were not considered

Reviewer 3 Report

In the manuscript “Self-powered bioelectrochemical nutrient recovery for fertilizer generation from human urine” the authors describe a novel design of a bioelectrochemical system to recover nutrients from urine. I consider that the authors made the necessary changes to make the manuscript clearer to the readers. For this reason I consider that the manuscript can be accepted for publication.